# Professional Narratives about Older Adults and Health Services Responsive to Fall-Inducing Frailty

**DOI:** 10.3390/ijerph20216975

**Published:** 2023-10-25

**Authors:** Laudicéia Noronha Xavier, Vânia Barbosa do Nascimento

**Affiliations:** 1Doctoral Program in Health Sciences at Centro Universitário Faculdade de Medicina do ABC (FMABC), Santo André 09060870, SP, Brazil; 2Department of Public Health, Centro Universitário Faculdade de Medicina do ABC (FMABC), Santo André 09060870, SP, Brazil

**Keywords:** aging, Primary Health Care, postural balance

## Abstract

The second external cause of death from unintentional injuries is falls in people over 60 and is a worldwide Public Health problem. Associated factors are identified early in Primary Health Care. Thus, we analyze professional narratives about older adults/old age and the organization of services in the presence of fall-inducing frailty. A structured narrative was applied under the following stages: understanding the context, setting/plot/character analysis, and interpretive synthesis. Data were collected from August to November 2022, distributing 21 health professionals in three Narrative Focus Groups. In the analyses, the collective conceptions dialogued with Bourdieu’s Epistemology of field, habitus, and capital. Technical and common sense representations of older adults were simultaneously observed among the results, along with the belief of old age as a problematic life stage. Care is centered on the installed disease/ailment. Encouraging autonomy and self-care emerges in integrative health practices, which older adults underestimate. Professionals access the lives of older adults according to their habitus, which, in turn, is structured (structuring) in the disputes for installed capital. Thus, the care provided disregards subjectivities and symbolic systems associated with falls.

## 1. Introduction

Individuals aged 60 or over are considered older adults under the primary documents in force in the country, such as the Report of the World Health Organization (WHO) [1], Federal Law 8.842/94 [2], Law 10.741/2003 [3], and other documents that update the topic or that are conscientious.

The WHO [1] points out that this older adult age threshold must be closely related to the socioeconomic conditions of each nation (and, therefore, the estimated life span) in such a way as to designate the age of 60 for developing countries, extending to 65 years (more) in the case of people living in developed countries [4].

A systemic growth in size, structure, and distribution of the elderly population is observed vis à vis other age groups. In an article [5], José Eustáquio Diniz Alves, Ph.D. in Demography, declares that never before has the history of humanity registered such record figures and that, from this point onwards, the total number of older adults worldwide will only continue to increase.

The new demography also triggers an epidemiological transition characterized by particular patterns of morbimortality, placing events of concern at the center, such as typical disabilities associated with aging, geriatric syndromes, heart and chronic degenerative diseases, and falls from standing height [6,7,8]. Significant here are the pandemic events that have disproportionately affected older adults and exacerbated any logic attributed to the demographic/epidemiological transition by how the needs of this segment of the population are responded to, which require changes [9,10].

From this viewpoint, researchers Jesus et al. [11] wrote about the greater frailty of older adults given the context of social vulnerability, suggesting that social assistance and health care are highly demanded in old age in the presence of vulnerability, which is unlike any other life stage. The term frailty is used as a synonym for older adults, which is why we seek to explain the functional decline and instability that trigger events harmful to health, such as falls [12,13,14,15].

Especially when people who age experience their biological decline in poverty, difficulty in accessing information and services, and suffering from other logics of non-inclusive societies [16], an outlook of significant vulnerability emerges among older adults living in impoverished communities [13]. This condition profoundly marks the city of Caucaia [17] in the Brazilian Northeast—the primary locus of the present research.

Etymologically speaking, the word “frail” is associated with the extreme weakness of a considered whole, referring to what is easily breakable. The cultural universe is designed so that vulnerable older adults can be assumed frail or at risk of “breaking”, becoming the target of hasty evaluations in the sense of diagnoses and consequent prescription of their guardianship by specialized medical services [13]. 

The word can have this cultural weight, prompting multidisciplinary teams to think and interpret the totality of being an older adult by solely regarding the affected part. The part is a specific condition of existence, such as the use and association of medications, a diagnosed chronic illness, or the physiological condition of loss of muscle mass or visual acuity. This assumption leads to the perception (i.e., the older adult is convinced) of frail health as a feature of someone who is also fragile, incapable, or completely impeded from achieving self-preservation or self-care and dependent on “well-intentioned caring actions” (p. 181) [18].

In many ways, being fragile (or robust) in relation to health can signal only one part of the older adult’s existence. The concept of vulnerability comes into play, as it brings the perspective or the need to glimpse, with and despite weaknesses, at the individual contexts that inform the availability/lack of support or overcome conditions associated with economic and educational markers, those of ethnicity and gender, besides other social injunctions [11], which determine individual or group’s abilities/power to resist (resilience).

The present study focuses on older adults under their dichotomous nature, in which they can simultaneously be/perceive themselves as capable and vulnerable—vulnerability due to the non-availability of diverse social support networks, translated into the combination of the need for access to information, material resources and others necessary to face cultural barriers or violent diktats [19]. As an affected social dimension, vulnerability does not ignore the older adult’s investment in other dimensions or intact personal abilities. On the contrary, the permanent exercise of overcoming several weaknesses is clear.

There is a tendency to consider that the life context of the vulnerable exacerbates their frailties, adversely affecting human aging [20]. Thus, falls among the oldest old are considered a true clinical-geriatric syndrome of a multifaceted origin [15,21]

The event of falls in older adults is described in the field as one of the greatest frailties of this age group. Reasons for this understanding regarding social harm and family dynamics [22] are listed. Moreover, the effects associated with prolonged hospitalization stays, with a very high risk of death, are of great relevance and interest, as falls have been classified as a global Public Health problem [23].

We should underscore that one-third of older Brazilian adults fall at least once a year [14,24]. The prevalence of falls tends to increase by around 50% [25] with age, with the highest prevalence at the age of 80. The risk of falling again is around 60 and 70% among older adults who have already suffered a fall, and approximately 20% of this population with a history of falls experience a fatal fall [26], which has been the object of interest of several studies [27,28,29] with different approaches [15,30,31].

The most prevalent health concern among Brazilian older adults is related to falls involving fractures, followed by total or partial physical disability [32], requiring the closer monitoring of the greater technological density of the country’s hierarchical network of health services [33].

### Demographic Transition and the Sociological Analysis of Realities 

The demographic transition is a global event. It evidences the ongoing aging of populations [34]. Save for still possible adjustments, the 2022 Census recorded a population of 203,062,512 in Brazil, and people under 30 declined by 5.4% from 2012 to 2021, while the growth curve for the oldest old is very intense. The share of people aged 60 or over hiked from 11.3% to 14.7% in the same period, with 31.2 million in the last census [35], implying enormous social challenges [5].

The demographic change has many demands but mainly requires an understanding of the aging process [36]. Besides the prevalent diseases in older adults, facts intrinsic to the illness inflict molecular and cellular damage associated with the gradual loss of physiological reserves and the natural decline in the general capacities of this population to defend themselves against external agents, increasing the risks for all types and causes of accidents, including falls from standing height (FSHs) [27,37,38,39,40].

Clinical problems and social and economic adversities accompany the national population’s aging. These ailments aggravate general life conditions, reducing the probability of obtaining help [41]. The consensus refers to the longevity framework closely associated with the demand for healthier spaces, better quality of life, and broad access to life protection services in all their scope [19,41,42,43].

In Brazil, healthy aging actions provide for comprehensive care, primarily provided by Primary Health Care (PHC) and articulated with care networks, respecting principles/guidelines of the Brazilian Unified Health System (SUS), without, however, underestimating vulnerable older adults’ life contexts and existence conditions [44]. The Brazilian Ministry of Health (MS) defends this premise through the Healthcare Secretariat, Department of Programmatic and Strategic Actions, which guides the implementation of the Care Line for Comprehensive Health Care for Older Adults in the SUS. The body recognizes that the issue is a priority and declares the need to mitigate the unequal opportunities, establishing strategies for better implementation and adherence in monitoring functional ability to offer autonomous and healthy aging [45].

In the same vein, the United Nations understands that the health care sector is the most demanded, and that there is an urgent need to adapt services and hire qualified professionals to work in primary care, health promotion, and comprehensive care for older adults. A specific UN agenda works toward reducing health care inequalities on four strategic fronts: changing the ways of feeling/thinking about age and how social segments operate vis à vis older adults; promoting the skills of older people; offering older-adult-responsive PHC services; and enabling access to (integrated/quality) long-term care to meet older adults’ demand [46].

Accordingly, gerontology (and social gerontology) is a multidisciplinary science that studies longevity in all dimensions, and values and considers qualitative interactions in families and communities as a pillar of healthy aging [47]. Thus, efforts based on institutional concepts have been made for solving public problems more aligned with these movements, i.e., toward sensitizing the world, achieving social and economic improvements, and, consequently, establishing the health indicators specific to the places where people in the age groups considered older adults live or relocate [48].

In this context, in its General Assembly report aimed at building inclusive societies for older people, entitled “The United Nations Decade of Healthy Ageing (2021–2030)”, the UN called for global and multisectoral collaboration towards concrete actions that promote sustainable, healthy, and long lives [46].

What is effectively intended from the health sector are care structures and practices (re)oriented towards the close and realistic monitoring of the growing demand caused by the ever-increasing amount of people living longer without necessarily being able to live physically, socially, and economically better [49,50]. In this context, Primary Health Care (PHC) is particularly crucial given its essential role in mapping risk and vulnerability situations, strengthening/benefiting households and populations in the localities, minimizing existing intercurrences and articulating new targeted public policies, and implementing the already existing apparatus [51,52].

The above justifies conducting a survey based on three problem questions: what do primary care health professionals feel/think about “old men” and old age? How do services act/organize themselves regarding vulnerable older adults? From the perspective of responsive health services, what actions to promote people’s skills apply to senescent individuals at risk of fall-inducing frailty?

To answer the questions in this article, we understand that PHC particularizes a specific field of economic, intellectual, and symbolic disputes and that, above all, this field does not exist without a habitus establishing it as such (and vice versa). Pierre Bourdieu [53] argues that a field is evidenced by the object of dispute and the power relationship between the stakeholders (or institutions) engaged in the struggle. Thus, PHC is a social and symbolic space, structured in and structuring the SUS, with its autonomous arrangement against other fields in which health equipment is installed. Bourdieu and his science of society [54,55,56,57,58] provides the best tradition of questioning the reason for a given configuration of the field, bringing a “rationale” of its functioning into people’s lives, but which is not at the same time “logical” for the culture of the natural stakeholders of the place where the struggles are fought, which the author calls habitus.

Regarding practices, “ideas” have interpretative potential, hence the wide acceptance/use of Bourdieu’s concepts in different knowledge domains about contexts and societies [59]. He thus idealized his concept of “field” to generalize social spaces, that is, to make this abstract social world, and made, at first, of representations that are recognized and studied.

The (economic, cultural, social, and symbolic) capital allows for the unique structuring of the field as a social space. Somehow, it relates to authority and power. As for the concept of habitus, Bourdieu says that it both is structured from and structures the social field. A body in this social field is historically contextualized so that it can be structured from the “inculcation” of culturally arbitrary establishments and events (family, school, church, and political parties), which also means that the past is in the stakeholders’ habitus, surviving in a non-crystallized form because it can continuously be molded [57].

This writing reflects the specific objectives: to analyze professional narratives about older people/old age; to address the primacy of responsive health services in promoting the abilities of older people; and to the reveal structural and cultural aspects of the vulnerability of older adults at risk of frailty-inducing falls from standing height.

Research on this theme is relevant due to the motivation to give voice to health care stakeholders for older adults, contributing to possibility of potential structural, cultural, and intrinsic barriers to professionals inherent in the set of conforming PHC actions for older adults, and to disseminate findings with the potential to add to the source of references that sustain the organization of practices and public policies to achieve the goal of promoting old age with independence, autonomy, and quality of life.

## 2. Material and Methods

We adopted the quality criteria from the COREQ (Consolidated Criteria for Reporting Qualitative Research) and SRQR (Standards for Reporting Qualitative Research) checklists. They are used as supporting tools to ensure transparency and completeness of qualitative articles. While the SRQR allowed for the evaluation of the best synthesis for the results and overall report quality, COREQ describes the focus groups’ field, recruitment, and implementation stages [60,61].

### 2.1. Study Design

This exploratory descriptive study emphasizes a qualitative approach [62] with contributions from Pierre Bourdieu’s Social Theory, in which the language of “stakeholders” informs the study participants. 

The proposed conversation topics allowed us to sensitize professionals to produce narratives close to the research objective. Data triangulation was considered to articulate field strategies, data collection instruments, processing techniques, and theoretical frameworks for analysis [62,63].

### 2.2. Sampling Procedure and Data Collection and Processing 

Primary Care in Caucaia, a metropolitan area of Fortaleza, Ceará state, Brazil, is organized into six districts with 80 Family Health Strategy (ESF) teams, corresponding to an 86.96% coverage of the population. The research was conducted in two Primary Health Care Units (UAPSs), also known as Basic Health Units (UBSs) or popularly called health posts. The UAPSs defined for this research are located in districts IV and V of the municipality. They were selected because they have a combined population of 22,855 people, 22.5% of whom are older adults [64,65].

The field phase occurred from August to November of 2022, considering the different stages in which access to spaces and data collection were given to 21 health professionals. 

We adopted purposeful sampling. Instead of “representativeness” [62], recruiting these participants prioritized a driving diversity of insights within the group/population under study. The research’s eligibility criteria [62] were higher education professionals with experience in PHC, one year or more of practice, and working in direct care for older adults during the research period.

When selecting health professionals, we attempted to hear the voice of the “other” with legitimacy to manifest their unique world, thereby unveiling their representations [66].

A variation of the Narrative Focus Group interview technique was used when collecting this type of indicator—an innovation based on participatory research designs [67,68,69,70,71,72]

The literature is very divergent regarding the number of people per discursive session [73], opting to divide the contingent of 21 participants into three groups of seven health professionals and adopting a 90 min meeting for each group. 

The inclusion criteria [70] for the composition of the groups did not consider gender, social class, ethnicity, or other sociodemographic variables. They focused instead on ensuring multidisciplinarity during segmentation in the FGs comprising health care professionals with higher education, with one year or more of PHC experience, and who directly provide care to older adults.

The portable recording device was placed strategically to capture the voices with the highest quality. The researcher assumed the role of moderating the debates, adopting a guide or roadmap mirroring the research questions, and offered the same bases of discussion for the three groups according to the agreed rules: all should participate; speak one at a time; avoid dominating the discussion; and keep the answers within each topic proposed during the debate [74,75,76].

We should also mention that this phase of the data collection work in the field was concluded when it was possible to establish the investigated empirical framework, considering the criterion of the saturation of significant statements when the answers to the research questions were repeated [63]; that is, the data were sufficient to support the conclusions [67].

Verbatim transcription was preferred for recording and organizing the analytical corpus, i.e., faithfully recording in writing everything said [62]. The transcription of the interviews took 25 h, with unique processing in retrieving the original lines from the researcher’s “linguistic filter”, including avoiding or not allowing slang, dialects that could hinder understanding, noise that did not represent speech, and repetitions due to the inherent structure of the language. 

Two verification reviews were performed to ensure the reliability of the audio recordings, and acronyms were used identified the participants to protect their anonymity during the establishment of the corpus of the written text in the second verification. We opted to apply the letters PS to represent all health care professionals, followed immediately by a differentiating Arabic numeral so that the first identification received the label PS-01, with subsequent participants being labeled in an ascending sequence, up to PS-21, per the order of the initial statement of each participant.

The corpus was processed for analysis according to the Gomes theoretical–methodological framework [77,78,79,80]. In Gomes’ proposal [80], narrative analysis is a method in itself. It was developed to unveil meanings from the world of health care and people’s experiences with illness. Therefore, organizing contextualized data around the sociocultural elaborations of disease and health is typical.

In this method, it is crucial to work with meaningful statements [66]. The analysis text is narrated in the third person, and the researcher is the “observer-narrator” who (re)tells and intervenes in the stories of the narrators-stakeholders (research subjects) [80].

Interventions should not signify the personal opinion of the narrator-observer. Thus, the interpretive lens or reading key is indicated when inferring [63,72]. From this perspective, the interpretative syntheses were made through the possible dialogue between the organized narratives and the main concepts formulated by Pierre Bourdieu (field, habitus, and “capital”), complementing the understanding of the context by the various authors who address the theme.

### 2.3. Ethical Aspects

Risks were minimized, benefits were reported, and the research followed the process of its ethical implementation protocols under specific parameters [81,82], with authorization from the Research Ethics Committee (CEP) of the Terra Nordeste-Fatene University to enter the field and implement the project. The protocol was registered under N° 5.752.849/2020.

## 3. Results

Significant information was collected on the FG participants, considering the present study’s objectives, as shown in Table 1.

The 21 health professionals who formed the focus groups consisted of two doctors, nine nurses, two physical educators, two physiotherapists, two nutritionists, two social workers, and two psychologists. In total, 17 were female and four male, with a predominant age group of 20–30 years. Regarding titles, eight mentioned specialization, and two mentioned enrolling in a complementary elderly health training course while obtaining their master’s degree. 

Below are the results supported by the themes that originated the conversations, referring to each issue-problem: the meanings about being old and aging triggered by the professionals’ narratives; the organization of work at the UAPS related to vulnerable older adults; and health services responsive to risk factors for falls among older adults.

### 3.1. Being Old and Aging—Senses Triggered by Narratives

As a prompt to the FG, the professionals were invited to talk about their previous knowledge, everything that was understood or experienced, and that made them create or recreate representations. In this regard, the following narratives are retrieved:

Old age is a natural course of human beings. The person is born, becomes a child, teenager, young person… Until old age. It is a stage of development with its specificities, and within society, this stage receives much attention from the health field. Many academic works focus on this phase of life. (PS-06); In graduation, at least in nursing, we see the stages of life a lot, and in one of these stages, we have disciplines on elderly health. In these, we see a lot about the diseases of this stage of life. We don’t usually see or study healthy older adults. So, the graduation course focuses on treating pathologies in older adults. (PS-03).

We should say that one narrative influenced another in group dynamics, shaping representations and favoring or emphasizing plots whose vocabulary was technical/academic or those that referred to times during higher education experience, in particular, to what they learned about older adults through specialized literature, referring to the representation of old age as a pathology.

As they kept talking and listening, the professionals tended to shift the topic of the discourse and, in the narrative excerpts below, signal that they will detach themselves from an academic perspective to delve into common sense representations—which is what this study pursues:

About old age… When you look and listen to the typical person talks, you hear that the old ‘is no longer good for much’. Now, as a physiotherapist—and I already have a perspective that may differ from other professionals—I learned not to look at older adults as if they ‘were no longer any good’. I was taught to look at his capabilities and work with this. (PS-08); I think that, to a certain extent, before graduation, the perception is very negative (referring to older adults) because old age already brings this nomenclature of old, outdated, that is no good any longer, that which you no longer use because it’s old and worn out. We work on this new perspective of old age and aging throughout our graduation in psychology. (PS-10).

In this regard, some health professions were able to produce favorable representations, especially those that understood the comprehensive being, while others, who were disease-focused, tended to maintain stigmatized representations. Similar responses were presented, and the highlight below shows how the narrators started to detach from this negative perception about older adults and old age, starting to infer it as a problem brought about by older adults into the therapeutic relationship:

What I see in my office is that, for many patients, aging is synonymous with suffering. I see a feeling of hopelessness. I see this older adult with very negative statements: “My life doesn’t matter anymore, my children don’t care about me anymore, and society doesn’t care… And I’ve done everything I had to do”. It’s as if, for him, life is already over. (PS-18); There is a considerable stigma related to older adults and old age. It (referring to old age) is considered negative […]. Because people don’t want to reach that age, and it’s because they feel the weight of negativity that accompanies representations: becoming dependent on other people, start moving less… (PS-12).

From their statements, we interpret that the common sense representation overwhelms older adults with its negative burden because it conforms to thoughts of elderly patients with some health impairment. That being the case, when formulated critically and reflectively, the representation expressed by the health care professional will likely help the patient reframe this harmful self-representation. In the highlight sections below, the nuance modifies or reclassifies the narrative under the protection of social slogans:

Old age is a stage that could be healthy, which could be the “best age”… As they say. However, in my practical experience, I see that little (PS-18); It is called the ‘best age’ but is rarely perceived that way […] (PS-19).

This mode of representation is said to conceal real/personal feelings; that is, one is facing (preconceived) understandings that were consciously or unconsciously inserted in the narratives [66,83]. Furthermore, the content of this representation denotes a belief that there is greater potency/power of other age groups in relation to old age, so being old is understood as a disadvantage from this viewpoint.

### 3.2. Perception of PHC and Organization of Work Related to Vulnerable Older Adults

Due to this classification system, the data block related to the PHC services is available, including how they are organized and the perception of effectiveness when considering the older adults’ demands. The analysis follows the logic of the order of data produced according to the stimuli: talk about the professions in place and that involved in care; address the available services/resources; and identify which strategies are used in health education and promotion”.

We are a multidisciplinary team with different emphases. So, each emphasis contributes according to its specificity in elderly care. We organize ourselves, therefore, to perceive their demands in a way that each professional category will have a perspective and, through listening, will perceive the main demands that each professional participates in (PS-11); It is very enriching and advantageous to have a whole team to assist older adults because they (older adults) listen to various subjects, and they learn, absorb, and pass on to their families and neighbors. Furthermore, like it or not, we bond with that user, and the user starts to trust the professional more. When guidance stems from a fine-tuned professional with the older adult (the bond), they absorb it and start to practice what they were taught to do (PS-04); I’m from the multidisciplinary residency. We can see the importance of each knowledge and how much knowledge complements each other. There is no way to talk only about nutrition with older adults with a whole life behind them… It’s not just food. So, we complement each other a lot. That is why the multidisciplinary residency program is so essential (PS-15).

When narrating about the modality of multidisciplinary care, the agents recalled the importance of complementarity of knowledge, besides mentioning some of the pillars of the therapeutic relationship, such as creating bonds and establishing a relationship of trust between professionals and older adults. However, only the former narrated about listening to older adults, whereas the vector is the authority of health care professionals toward older adults in the others. Moreover, FG workers held a uniform type of discourse regarding the teams’ purpose. They continued to expose the most known/disclosed design of multidisciplinary practices and the meanings attributed to the need and reason for the teams’ existence. Again, regarding the organization of their work with older adults, the workers remembered their respective actions:

I also work in the guidance of rights. Sometimes, the older adult is sick and suffers from property violence, sometimes without even knowing it. It is violence where older adults do not access their money, although they are oriented/aware. We also have moral violence and negligence—even negligence that occurs in a specific institution and, therefore, they need to know their rights and where to claim them. (PS-07); My professional knowledge can sometimes help with the most essential nutrition information. So, I believe that the basics that I talk about food already interfere a lot with their health and diet… They start to understand. (PS-08).

From a somewhat myopic perspective of health care professionals, no reference was made to the intersectoral network and the cross-sectoral approach to integrating knowledge/experiences for case management [84]. Thus, the statements reflected the services in the better-known and consolidated format: fixed structures, local stakeholders, and strict care protocols. Only two narratives come closer to describing what human complexity is in primary care:

Regarding nursing, you end up seeing everything. That is, you will be able to access more information about that patient more and more in its most comprehensive form as you interact with that patient: the social issue, the issue of medication use, diseases, family history, the relationship with the partner, anyway. […] We gradually realize these things with the office practice or bond with the family/patient. You try to clarify and start directing the care to the specificities of a patient… Furthermore, understanding that one person’s need is not another’s, right? (PS-12); In primary care—especially when working with the same community for a long time, serving the same population and families—you can look at a family core. It is not just looking at that patient sitting there, but you remember that the day before, you served the pregnant granddaughter of that older woman. You know that the granddaughter lives with her… So, you will understand how the network works, which even favors your understanding of that patient’s family and social context. […] As you exercise this knowledge in your clinical practice, you become more and more prepared to identify certain issues and (try) to solve them. (PS-06).

We should remember that the field occupied by PHC in its larger Family Health Strategy (ESF) proposes developing the work to promote ordinary people to the condition of reading realities so that they can assess what is best for them. They decide based on the best consensus (considering personal and community resources), and that is what autonomy is about, whose most evident strategy is encouraging people’s self-care in health. However, this was only heard of now. Continuing with the FG, in the stimulus given to address the strategies used in education and health promotion, we mention the following examples of narratives:

Primary care should have a lot of this prevention and health promotion perspective… (But) It doesn’t have much of a preventive perspective. (PS-14); This health promotion and prevention issue is in the background, and we serve a tiny audience of older adults for this purpose. (PS-17); It’s hard for us to find someone here, an older adult who comes here purely for health promotion. I see more already in the part where we have to solve something. (PS-20).

Derived from these narratives was the insight for the FG to address older adults’ representations regarding PHC spaces. The narratives produced revealed that besides the PHC system not remembering to use intersectoriality and partnerships with other social equipment in the region to offer health promotion activities for older adults, its professionals will complain that they should know/aid in prevention or move to seek help with little or superficial understanding of the condition of vulnerability to which older adults are exposed.

Most people seek the Health Center more often when they already have a health problem. It’s more about medication. […] the hypertensive, the diabetic… who come more for medical appointments. (PS-14); In primary care, older adults seek the service more from a more curative perspective, more for recovery when they already have a health problem. (PS-17); When he comes, he often comes with depression, anxiety, and dependence on psychiatric medications. (PS-18); I’ve been following that, increasingly, people look for the PHC Unit, the health service, when they already have something acute. It’s much more problematic. When people get here, it’s already an emergency. So, those who are followed up here only seek the health center when they have urgent needs. Then, unfortunately, we have to work on this demand, always trying to guide. Sometimes, it even frustrates us because it’s almost as if we were trying to solve something that wasn’t in the past. (PS-20).

The perception is that patients “do not perform” the promotional/preventative actions and only seek PHC when “they are already sick”. Still, in the health care spaces, the patient, i.e., the older adult, is seen as someone accumulating doubts, who is unaware or does not know (about his illness and care and medication use), needing the health professional to behave better.

Next, the narratives added the “ingredient” of vulnerability by talking about older adults’ lack of formal education, local violence, and other risk indicators for illness. Everything comes to the detriment of the more coherent frequency routine of older adults to health services:

Due to a lack of formal knowledge (I think this is a striking factor in this population), these people often do not know that they need to undergo a routine examination; they do not know how to relate a headache to high blood pressure… Thus, they end up in the emergency room. (PS-07); aging here is more complicated due to vulnerability. I think that’s it: a tiresome and suffering aging, full of violence; therefore, more passive aging in self-care, less active in the presence of support groups… (PS-11) Due to social conditions, society and the family do not offer support… So, they only come to the health center when they already have something chronic. (PS-12).

Regarding health service shortcomings, we should note two particularly revealing narratives:

We have from very active older adults […] to the bedridden patients who cannot come to the unit. We have those who come and report a fall: they fell in the street, on the way to the doctor’s office… Or they complain of some other problem. We access and intervene as the patient shows himself to us as he talks about his difficulty. (PS-06); That time and the number of people we serve do not allow us to sit down and discuss everything with them. We attend 17 in the morning and 17 in the afternoon, plus triage. We need to find a way to sit and talk about everything. Our time is rushing. Furthermore, it’s not getting any better. (PS-21)

With these narratives, professionals exposed the prevalence of problem-based care and make clear their working conditions and demands, showing that they also need to address their connection and limitations. The narratives of professionals who addressed the theme of older adults in their relationship with their families remember that every family lives the reality of the vulnerable. The constructed plot was that older adults are found in the settings (homes and communities) taking care of relatives, children, and grandchildren, always with a bias towards family contexts that have fewer partners and are more incomprehensible:

Older adults are there supporting the family financially and with food. They support raising their grandchildren, too—they take care of them so their children can work and have a productive life. (PS-08); Sometimes, older adults are the only providers in the family, and they sacrifice/neglect their care. (PS-09); Some older adults are overloaded: they must cook food for their children and grandchildren. Sometimes, other relatives come in for lunch. They feel responsible for this… (PS-12); Some older adults are there with a child on drugs, and they must put up with it. (PS-20).

Besides the older adult issue, what transpires in these narratives is a sick social system with no perspective and little capacity for self-elaboration. Perceptions about the low presence of men (and male older adults) looking for care will be elucidated in other narratives:

I attend to older adults and see a gender bias: women look out for me. (PS-12); It is infrequent to see a man. […] They only come to seek treatment when the disease is already settled (PS-13); Men voluntarily seeking the post is almost nonexistent and is very rare indeed. (PS-19).

According to surveys [55,61], women (including older women) are concerned with general care practices for several reasons, but mainly for resolving the health of other family members. They also accept invasive exams better and are more responsive to the biomedical discourse as a device for controlling “the sick body”. Whether or not this is recognized as cultural, it is still noted by professionals. This includes gender bias, ethnicity, social class, and education as variables that deserve attention from PHC professionals: Within the range of women I see, we should add that they are also poor, less educated Black women. (PS-13).

### 3.3. Responsive Health Services and Risk of Falls among Older Adults

The following narrative is highlighted regarding the factors associated with systemic imbalance with risk of falls among older adults:

The problem in old age begins with motor issues and walking alone. Because some will have Parkinson’s, others may get diseases in their hands, including Chikungunya—many are in much pain, unable to do simple manual activities. We have motor difficulties, the issue of the senses (vision, hearing, smell). Sensitivity is reduced in all of this with advanced age. So, to chew, they lack teeth; vision is lacking to see things well; hearing is also poor. Walking alone is complicated: they don’t hear a car horn or may not feel other dangers. If they fall, they have other problems walking, moving, and preventing them from falling again. (PS-17).

When it comes to falls, the architectural factor of the city was recognized along with others that make sense when addressing the affected systemic balance, including socioeconomic factors:

The architectural issue, the city, the place where he lives, and the issue of sidewalks. All this has to be considered when talking about falls in older adults. The path that they take to solve their activities is difficult to access. This will interfere with possible falls. (PS-07); These older adults are not taking care of themselves. They don’t exercise. They don’t eat right—they are eating poorly. Many of them are eating instant noodles and junk food. (PS-08); Lack of food or poor quality can cause weakness, muscle pain, inflammation, and even increased blood pressure. (PS-20).

According to the statements, it is possible to see the narrators’ reaffirmation of old age as a phase of life that involves the management of cognitive and physical weaknesses/limitations in the face of their environment, food and economic difficulties/deficiencies, and complex family relationships. Other representations came as voices from the field: they are professionals with home-visiting practices and will approach the theme from the perspective of this universe:

Numerous factors interfere with balance and the possibility of falls in older adults. When visiting the family, we have to see the physical and human structures inside the home: if the home has a built-up area at the bottom and the top; if there are any rear areas (backyard); how many people reside; if we have children who will leave a toy on the floor; whether it has bathrooms with facilities; anyway… Sometimes, older adults do not control their blood pressure with medications. They only take them when their pressure increases and they still eat inadequate things—according to them, because ‘the money is not enough’. (PS-20); Older adults at home may slip and fall in the bathroom. Even food, because not eating well can make you feel dizzy and have an imbalance. The use of medications is also a problem. With the increase in anxiety and depressive disorders, patients start taking a little medication to sleep and become number, which can contribute to that. (PS-07).

Among the central points highlighted as risks of falling are “non-adapted” housing conditions for older adults, the intergenerational “care overlap” of grandchildren, and the side effects of the concomitant use of medications. Regarding primary health services responsive to older adults, therapeutic groups were recognized as strategic actions to determine “healthy aging”. As the stakeholders-narrators point out:

The issue of healthy aging… I want to mention the issue of groups held to keep balance in older adults and avoid other complications. The CRAS (referring to the Social Assistance Reference Center) and us here have physical activity work. In short, this type of more bodywork and health education can be done to help prevent it. (PS-13); In socialization groups, we usually do many health education actions besides interprofessional appointments, where we help each other and complement each other to observe this older adult holistically. (PS-12).

Thus, the workers’ FG were urged to address interdisciplinary interventions and integrated therapeutic plans from the perspective of institutions responsive to healthy aging:

Older adults come for more than just lectures here, but for therapies: Reiki, auriculotherapy… They manage to relieve their anxiety. […] It’s even a way to leave the family environment, where there is so much stress, to arrive in an environment where they can talk, feel welcomed, and be well cared for. (PS-05); We have a group here called Healthy Life. Older adults feel livelier in this group, with the power to speak and use their widespread knowledge… This group gives them a voice and nurtures their social and community relationship. (PS-08); For example, we have projects here: Pilates and yoga for ladies, which greatly help. They won’t cure, but they will help somewhere. (PS-12); Some do Pilates groups, and others also participate in NASF older adults’ groups or attend CRAS elderly groups… So, many have this engagement spirit (PS-20).

We note the narratives that are suggestive of integrative and complementary health practices in PHC, suggesting an openness to understanding older adults in their entirety, including considering the context of receptive coexistence so that they can express themselves in a safe environment.

The PHC philosophy itself provides for the achievement of some independence and autonomy for older adults by promoting the inclusion of these people in therapeutic groups or unconventional practices that are characterized by being distinguished from biomedicine by not focusing on medication, and as being low-cost, integrated with family and society, and putting the subject in an active position regarding their own care [85].

## 4. Discussion 

Bourdieu’s General Theory of Fields [86]—since the analysis of health events still lacks theories from the social sciences—mainly applied the Gomes’ analysis method [77,78,79,80] and underpinned the type of dialog/debate of the narratives, thereby safely addressing the theme of aging, making clear the intellectual capital for each professional within the PHC [87]. It should be stated that local disputes stand behind all intellectual capital, which is expressed by the offices and positions of power that demarcate the field in which the hierarchy of authorities/power is made explicit [88].

In their narratives about how they feel and what they think about older adults, at first, the professionals speak based on what they learned in their disciplines and in the training courses, and for this reason, the type of narrative established is impersonal and precautionary against setbacks of any expression likely to offend, exclude, or marginalize this group of people. We understand that the universe of formal learning allows stakeholders to safely address the issue of aging while making clear the intellectual capital set for each one, also delimiting respective powers (or lack thereof).

However, it was necessary to overcome this FG phase, encouraging the narratives to migrate to subject (or group) conceptions about the elderly and old age that were closer to the social relationships investigated; the statements had to reflect the set of opinions/values of the stakeholder-narrators who, in turn, determine their conduct/behaviors vis à vis older adults. Narratives about the elderly and old age that are closer to the investigated social relationships emerged from the stimulus to the group. That is, the narratives in which old age is referred to as a bad phase of life emerge when people lose their usefulness for the family and social life. There seems to be (consciously or unconsciously) a perception accompanied by the stigma that being old is not good or that those who are old are “no longer good for anything” in such a way that older people may prefer not to identify with the old or will try to hide the signs of aging in every way. In the narrated PHC setting, it can be said that this habitus structures (and is structured by) a mental mode of representation, perpetuating non-positive practices toward promoting skills in senescence.

From the proceedings, we should remember that elaborating an identity involves different sociability processes, with social self-elaboration [89] serving as an example, and that all health care workers should be concerned with any thoughts that consider older adults as “no good”.

Furthermore, the perspective of a health service responsive to the frailty of older adults also involves the sustainable promotion of changes in the ways of thinking about age, including taking prejudicial actions upon the older adult [46]. In order to debate this topic, we should remember that people have already lost almost everything they could capitalize on in terms of power (job, income, possessions, relationships, and socially recognized knowledge) in the older adult life phase, which will condition their lifestyle and opportunities [53]. This should be included in the structured (and structuring) plan of the necessary social changes.

The narrator-stakeholder also stood out due to statements about old age, such as “it’s called the best age, but it’s not perceived that way” and “it could be healthy, but I don’t see it that much”. At this point, we observed contradictions, as evidenced by affirming something that was denied in the same argument. When contradictions or oppositions occur in a discourse, linguists say that it (the contradiction) will be seen as a failure. Thus, a principle that governs the choice, vision, and representation of the world under a tendency of this type of conclusion (so-called monologic) is the expression of one’s thoughts aloud [83]. Therefore, for this research, it is crucial to identify the narrators: who are they?

Regarding the variables listed during the characterization of the agents participating in this study (the category of health workers), in the composition of the narrative FG, young women, aged 20–30 and with an average of five years of training and a minimum of two years of seniority in the PHC prevailed. These data and the directions of the discussions lead us to assume a line of interpretation in which the entire set of statements on the agenda should be considered more for what the narrators hide and less for what they reveal; the tendency of this younger female professional is contingent upon prevailing in PHC elderly care, influencing the social order of this field, and implying a new attitude/perception of the sick body and interpersonal relationships since women are more culturally dedicated to caring for others.

On the other hand, if it is a young woman’s choice to train in a health field, it is not necessarily her aptitude nor even an option to start her work in assisting the human population of the older adult age group, which may trigger dissonances [90] with the gestalt that the care relationship facilitates.

This dissonance between personal values and work tasks is one of the leading causes affecting the mental health of health care professionals [91]. It stems from the stress of professionals believing that the work environment somehow threatens their status quo with excessive demands or that they (workers) do not identify with, or their cognition is contrary to, the impending demands [90].

Complementing this reasoning, we also see a growing number of young people from underprivileged households [92] taking higher education courses. Pierre Bourdieu understands that it is how the popular classes compensate for their disadvantage in the face of class culture [53].

Concerning the meanings attributed to the professions available and involved in assisting older adults, workers recognize and value the strategy of sharing cases and knowledge in multidisciplinary health care teams. The public policy prescribes multidisciplinary work for Primary Health Care [93,94], including visits to homes and communities—both the suburban environment when it comes to cities, and the most remote corners of access in the country—with the intention of solving the most frequent and relevant issues for the populations of the territories in which they settle [74]. They pursue indicators and variables that involve combating the emergence or deterioration of morbid cases, thus preventing populations from needing institutions of greater technological density in health care in which the care is depersonalized with high risks (in particular for older adults) and significantly increased costs for public coffers [45,93].

We take here, as a reference, the reading by Ecléa Bosi [95] in the context of the research and adapt it to the subject at hand in order to consider that professional interventions (and the experience of the care relationship) are all the more valid for older adults, as if the professional (observer) does not make “sanctuary excursions” in the situation of the older adult (observed). Collective Health [96] addresses the subject as “an honest dive” into the universe of clients, which in this study refers to (re)visiting the fields of PHC practices and the daily life of older adults with an ethical attitude to absorb their knowledge and customs as well as real-life conditions, and to participate in experiences, get closer to the local culture, create bonds, and consolidate an understanding of “which”, “how”, “with whom”, and “when” to solve or mitigate problems.

Women with a vulnerable profile living in Caucaia suffer daily from public disrespect and the most diverse forms of violence. It is the municipality with the highest occurrence of femicide in the State of Ceará [97]. We should ask the health services to mention what measures are taken towards empowering these women since the combination of factors and contours of structuring poverty produces oppressive contexts and a specific type of historical racism that is very intertwined with stereotypes, highlighting, of course, invisibility as its social capital.

However, returning to Bourdieu [98], when he addresses those “included in the field”, this author speaks of stakeholders as being favored individually by the volume of their capital (which is of the most diverse natures). In the name of this capital, these stakeholders tend to perpetuate dependent relationships to preserve the field, representing modes of domination: fixing stakeholders in a pattern, and creating and keeping the idea of a “need” for some and not others. Referring to PHC spaces, each stakeholder is taken by this logic intrinsic to the position/profession in such a way that all, under their intellectual capital, act to preserve their position or status in a field of practice. The more stakeholders that occupy a hierarchical space in the field, the greater their power, the more they will work (they will be willing to fight and exercise dominance and control) to create dependencies and tame (develop acceptance) others who occupy more disadvantaged positions of the structure, sustaining the chain of submission. This process is, to a certain extent, unconscious.

However, they also demarcate a change in the knowledge–power paradigm, with strategies such as therapeutic circles, which are foreseen to take place within PHC, as spaces for the democratization of knowledge, with users and professionals, in general, participating in them under the prerogative of the case study, formulating an integrated health care project [99,100,101,102]. The professionals in this study brought this theme to the FG when they addressed professions in PHC, adding potential and sharing knowledge within therapeutic groups and integrative/complementary health practices—the PICS [85].

Since the Alma-Ata declaration, one of the most significant references in Primary Health Care, Brazilian health-reform scientists have concentrated on the need to project the role of health teams [103]. The aforementioned SUS was established as a project that was not limited to a single sector, incorporating the ideas of integration, territoriality, and social participation to guarantee rights in the context of an expanded concept of health. Given the complexity of real life, one would assume that joint work with other areas and fields of knowledge would think about/implement changes influencing the causes of illness [104].

We should highlight the realities or examples narrated in the FG of this study. However, the stakeholders are within the conscious knowledge, finding the ways of thinking/organizing (each according to a particular symbolic capital) the Collective Health practices [105] in the more traditional and structured practices. Hence, the strategies mentioned for PHC (with the nature of help groups, therapeutic groups, or even alternative practices) prescribe actions of the same “formula” for different people, anchored on the assumption that “each” individual experiences similar situations and life content. Most stakeholder-narrators recognize these therapeutic groups, but, above all, think about the modus operandi prescribed for this health field, which is circumscribed by the PHC. Furthermore, the list of interventions listed by health professionals to work on promotion and prevention, and interventions intended to be performed within the older adults’ homes, tend to disregard, not listen to, or not consult this target population about their preferences, knowledge and practices, belief or symbolic systems, its weaknesses and vulnerabilities [106,107]. Therefore, professional practices can be understood as arbitrary, reflecting the limited scope or the identified/declared non-adherence.

Certain studies [99,100,101,102] discloses the fact that each profession in Primary Health Care, with its particular knowledge, underpins or forms the pillar of therapeutic plans or expanded projects of “interventions”, which insinuates that all knowledge is cross-cutting and interchangeable in these interventions [108,109]. From this perspective, most of the narratives that address “therapeutic groups” reinforce the characteristics that they originate from the sum of professional talents to diagnose risk and initiate cycles of promoting a healthy life or minimize harm associated with morbid events, anticipating their effects on the daily life activities of older adults. 

We should remember that PHC is part of the larger field of Collective Health, which, in turn, dates back to the founding principles of Social Medicine practices [99], which is considered to broaden the vision of older adults and to alleviate the weaknesses/vulnerabilities of this population group in the face of adverse events, with measures to monitor social and economic conditions so that they live with better quality, have healthy habits, and interact with social networks for more autonomy and independence, thereby reducing the biological harm that incapacitates older adults [18,106].

We consider that the structure (with all functions) of the culture of one field does not have to be reduced due to any universal (physical, biological, or spiritual) principle by the culture of another. There will be a field where older adults live, and not where UAPS professionals work, if the opposite occurs. This will also be a source of endless disputes, regarding which only the PHC as an entity would promote the encounter between the fields, making them (con)fused. 

Thus, the examples of strategies mentioned were designed to bring older adults into the service, with fixed facilities and scheduled times, service protocols, and an demarcated object that originates from each individualized perspective, that is, of the physical educator, the physiotherapist, the social worker, and the psychologist. In this way, aging for health professionals in the UAPS is defined by biofunctional and form changes [110]. While the functional capacity of older adults can be approached per the difficulties or even the lack of skills to perform certain activities of daily living. The morphology informs bodily changes, resulting in instability, decreased strength, and postural changes that affect gait and balance.

The systemic effects of biochemical aspects of aging are also implicated in mobility due to the neural regulation of the center of gravity and, among others, the reduction in bone and muscle mass [111]. Thus, fragility and vulnerability were associated with difficulties in accessing prevention and treatment services (in addition to the city’s architectural problems and economic, cultural, and physical/biological factors), the abandonment of older adults, the lack of family and social support (social/relational dimensions), and medication side effects.

The apparently rigid structure behind this behavior forces some to adapt to what is predestined for the health care field. In other words, care and actions are prescribed for the health of older adults. However, they do not necessarily consider all the conditions of the life dynamics of this specific population, with their intrinsic frailties, such as living in households with modern and diversified designs and, in particular, in communities that are more vulnerable to mobility (through alleys and bumpy streets), access to resources (scarce and in dispute), and physical risks (associated with parastatal power disputes) [97,112,113]—of which the Municipality of Caucaia is an example, not the exception, in the country.

At the very least, these statements bring potential reflections to subsidize the methods of necessary changes and (re)orientation of clinical work in Public Health aimed at older adults. Regarding the reproduced excerpts, one habitus function is to account for linking the practices and intangible consumer goods of the individual stakeholder or the group of stakeholders [114], and this is not precisely related to formal education but stems from previous experience. Thus, what happens with the problem of the consumer good represented by health is that it is more like a discourse in the moral-cultural scope than as a method of expressing concerns accompanied by practical actions by public power entities to transform it into possible mass consumption [52,93]—and the general population accepts this. Older adults who experience several other needs can also accumulate adverse experiences with health care in such a way that not proactively seeking a health post is just a hesitant behavior in the face of obstacles.

It is challenging to argue in favor of a Universal Health System in the context in which it is reduced to a mechanism for caring for less affluent people. It is not uncommon for health posts to experience problems (some of them chronic), such as lacking professionals, medicines, devices, and essential services. There needs to be more bureaucracy and a method to prevent significant delays in performing services. Jairnilson Paim thus speaks on the “risk of dismantling the SUS” when, in government policies, health is a peripheral issue [93].

The Narrative Focus Group achieved its objective of giving a voice to stakeholders and establishing an understanding of settings, plots, and characters. They also began to reflect on the mechanisms that move the PHC context while leaving clues of their perceptions of older adults, beliefs about the organization of services, and more ingrained practices.

As a limitation of the study, we should underscore that only four men participated in the sample, which could indicate a gender bias in the statements; this is due to the high degree of female participation in health higher education. The short time spent training professionals, coinciding with their length of experience with health care for older adults in PHC, is also a limitation.

We should point out the need for new (longitudinal) studies that also involve older adults to understand the effects of this growing trend of young women being attracted to health courses, graduating in this field, and starting to work with older adults in PHC without necessarily having an aptitude that defines their best practices. Thus, historical verification studies will also be very relevant to understanding the cultural aspects involved in creating and transforming the concept of health that governs practices.

## 5. Conclusions

The negative reification of the representation of older adults by most health professionals participating in the research is closely related to their understood, experienced, and projected self-representations. Instrumentalized by this bias, the stakeholder-narrators can perceive and make associations with the causes of the primary suffering of the elderly population, including concerning older adults’ needs to adapt to their new reality and demands of modern family arrangements. However, health services continue to focus on resolving “complaints”.

From the tradition of behaving and presenting themselves in response to walk-in demands, PHC professionals treat patients and illnesses. The offered therapeutic spaces (or groups) function as a formality and thus are perceived as a political locus for the control of all bodies and the submission of all lives, with little margin for moving stakeholders within the field given by the institutionalized field.

Although the field comprises many intellectualities (multidisciplinarity), primary care is not structured according to the stakeholders but rather encompasses all the stakeholders of that specified field. So, diseases related to longevity, the risk of falls, and other conditions that result in death are treated by stakeholders according to the incoming demand and in the context of routines and access protocols, approaches, and treatments. The health field is designed as disciplining spaces full of valuable and docile bodies to produce political responses to “diseases” as social ills.

## Figures and Tables

**Table 1 ijerph-20-06975-t001:** Distribution of data on the characteristics of research participants. September to October. Caucaia—CE, 2022.

Variables		Fa	%
Gender			
	Female	17	81
	Male	04	19
	Total	21	100
Age Group			
	20–30 years	11	52
	31–40 years	03	14
	41–50 years	05	24
	50 years and over	02	10
	Total	21	100
Professional Category			
	Doctor	02	10
	Nurse	09	40
	Physiotherapist	02	10
	Physical educator	02	10
	Social worker	02	10
	Psychologist	02	10
	Nutritionist	02	10
	Total	21	
Years of Study			
	0–5 years	11	52
	6–10 years	03	14
	11–15 years	04	20
	16–20 years	02	10
	21 years and over	01	04
	Total	21	
Title			
	Graduation	15	71
	Specialization	06	29
	Total	21	100%
Work Seniority in ESF			
	0–1 year	06	29
	1–2 years	07	34
	2–3 years	03	14
	3–4 years	03	14
	4–5 years	01	04
	5 years and over	01	04
	Total	21	100
Elderly Care Training			
	Yes	02	90.5
	No	19	9.5
	Total	21	100

**Source:** Own elaboration.

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
