# Peer review of "Professional Narratives about Older Adults and Health Services Responsive to Fall-Inducing Frailty"

_ijerph, 2023, doi:10.3390/ijerph20216975_

Round 1

Reviewer 1 Report

Comments and Suggestions for Authors

I am very pleased to have reviewed the manuscript entitled “Older adults and Fall-Inducing Frailty: Bourdieu’s analysis of field narrative”.

I have several comments:

INTRODUCTION

-          The introduction is very long. While the contents are interesting, this length makes it difficult to consider which elements are the most important. Some sections, such as point 1.3, may need to be shortened.

-          Line 113: it still seems to be talking about article number 3. In this case, the bibliographic reference should be changed, as it appears as number 16.

-          It is not clear what the objective of the study is. It is recommended to include the objective at the end of the introduction.

MATERIAL AND METHODS

-          This section is very long. There are parts describing, in general terms, methods used that could be shortened.

-          What criteria were used to allocate the participants (21) in the three focus groups?

RESULTS

-          It might be useful to include a table with sample data (lines 344-350).

-          Lines 350-2: this may be a limitation of the study. Better to include it in the "discussion" section. The professional experience of the focus group participants may also be a limitation.

-          Some paragraphs interpret results (lines 456-462; lines 524-530; lines 544-555; lines 556-561; lines 606-612; lines 666-672). They are more appropriately placed in the discussion section.

DISCUSSION

-          A section on the limitations of the study should be included.

-          Are there similar studies in the literature with which to compare?

-          It should be noted that in several points stakeholders are mentioned, but the work has focused only on health professionals. It would be interesting to include other actors, such as older people, their carers, etc.

CONCLUSIONS

-          The conclusions are very long. They should also be clearly related to the objectives of the study.

Author Response

We are grateful for all the help provided by the evaluator's opinion. We performed, to the best of our capabilities and understanding, the suggested changes for improvements, in the certainty that there will always remain viewpoints to be considered.

REVIEW FEEDBACK

Dear reviewer,

We are grateful for your relevant suggestions, reviewing the work entitled Older Adults and Fall-Inducing Frailty: Bourdieu’s Analysis of Field Narratives.

ACADEMIC RESPONSES

INTRODUCTION:

REVIEW: “ The introduction is very long. While the contents are interesting, this length makes it difficult to consider which elements are the most important. Some sections, such as point 1.3, may need to be shortened.  Line 113: it still seems to be talking about article number 3. In this case, the bibliographic reference should be changed, as it appears as number 16. It is not clear what the objective of the study is. It is recommended to include the objective at the end of the introduction.”

POINT-TO-POINT RESPONSES

Reviewer’s request/suggestion

Responder – Authors' response/adaptations

“The introduction is very long.”

We adjusted.

“Some sections, such as point 1.3, may need to be shortened.”

We adjusted.

MATERIAL AND METHODS:

REVIEW: “This section is very long. There are parts describing, in general terms, methods used that could be shortened. What criteria were used to allocate the participants (21) in the three focus groups?”

POINT-TO-POINT RESPONSES

Reviewer’s request/suggestion

Responder – Authors' response/adaptations

This section is very long. There are parts describing, in general terms, methods used that could be shortened.”

We adapted the style to match academic writing in this topic.

“What criteria were used to allocate the participants (21) in the three focus groups?”

Now, it was stipulated in the text body (see lines 232-236).

RESULTS:

REVIEW: “ It might be useful to include a table with sample data (lines 344-350).  Lines 350-2: this may be a limitation of the study. Better to include it in the "discussion" section. The professional experience of the focus group participants may also be a limitation. Some paragraphs interpret results (lines 456-462; lines 524-530; lines 544-555; lines 556-561; lines 606-612; lines 666-672). They are more appropriately placed in the discussion section.”

POINT-TO-POINT RESPONSES

“It might be useful to include a table with sample data (lines 344-350).”

We appreciate the guidance. We have inserted this quantitative data visualization feature (see lines 284-286).

“Lines 350-2: this may be a limitation of the study. Better to include it in the "discussion" section.”

We made the suggested inclusion (see lines 784-786).

“The professional experience of the focus group participants may also be a limitation.”

We considered this truth and made the necessary adjustments in the new wording of the article (see lines 786-788).

“Some paragraphs interpret results (lines 456-462; lines 524-530; lines 544-555; lines 556-561; lines 606-612; lines 666-672). They are more appropriately placed in the discussion section.”

We accepted your suggestion and moved the referred paragraphs to the interpretations chapter (see lines 668-674/695-701/709-716/725-731/762-773/774-779).

DISCUSSION AND CONCLUSION:

REVIEW: “A section on the limitations of the study should be included. Are there similar studies in the literature with which to compare? It should be noted that in several points stakeholders are mentioned, but the work has focused only on health professionals. It would be interesting to include other actors, such as older people, their caregivers, etc. The conclusions are very long. They should also be clearly related to the objectives of the study”.

POINT-TO-POINT RESPONSES

“A section on the limitations of the study should be included.”

We included paragraphs referring to the limitations of the study (see lines 784-788).

“Are there similar studies in the literature with which to compare?”

In Brazil, nursing science has shown a tendency to adopt interpretative references originating in Sociology (also) when assuming the objects: “primary health care”; and/or people in the stages of the life cycle, including the “older person”; and/or “risk of falls in older adults”; and/or “quality of life, independence and autonomy of older adults”… – including the possibility of adopting samples from health professionals and/or older adults; and/or family members and people caring for older adults; adopting the framework of Pierre Bourdieu's theory of social action as a key for interpretative readings; and/or narrative research design. It just happens that, in the study in question, all these elements are considered for the specific reality outlined. Some of the research identified was cited by us, others were not. Examples of generally accessed studies are:

da Silva, T. F., & David, H. M. S. L. (2020). Análise do campo da Atenção Básica à luz da teoria de Pierre Bourdieu. Revista de APS23(3).

da Silva, T. F., & David, H. M. S. L. (2018). O campo da atenção básica: uma reflexão epistemiológica pela lente de Pierre Bourdieu. Revista Sociais e Humanas31(3).

Neves, A. C. L. (2019). Estratégia Saúde da Família e pessoas com hipertensão e diabetes: redes sociais e longitudinalidade. Dissertação (Mestrado em Enfermagem) - Faculdade de Enfermagem, Universidade do Estado do Rio de Janeiro.

Caffé Filho, H. P., Vieira, D. D., de Oliveira, L. M. S. R., Beserra, E. A., & da Silva, A. M. C. F. (2021). Narrativas, Influências e Experiências: uma breve análise das reflexões de Pierre Bourdieu/Narratives, Influences and Experiences: a brief analysis of Pierre Bourdieu's reflections. ID on line. Revista de psicologia15(57), 104-112.

do Nascimento Serra, J. (2010). Violência simbólica contra os idosos: forma sigilosa e sutil de constrangimento. Revista de Políticas Públicas14(1), 95-102.

 “It should be noted that in several points stakeholders are mentioned, but the work has focused only on health professionals. It would be interesting to include other actors, such as older people, their caregivers, etc.”

We respect and appreciate the suggestion. Actually, the article discussed here is an excerpt from a larger thesis work, and interviews with older adults were indeed included. All material, however, exceeds the capacity (limited in number of pages or characters) of a single article.

“The conclusions are very long. They should also be clearly related to the objectives of the study.”

We reviewed this topic, ensuring conciseness and precision in reporting achievement of objectives.

Reviewer 2 Report

Comments and Suggestions for Authors

The research analyzed aims to: we analyze professional narratives about older adults/old age and the organization of services in the presence of falling-inducing frailty.

Considering the strengths and limitations of the analyzed research, the following questions are pointed out:

1)      There is no evidence of an acceptable relationship between the article title and the investigation objective, specifying the organization of the services.

2)      It is recommended to specify the improved research objective in the last paragraph of the introduction.

3)      The research is described as quantitative (Line: 238). However, all the descriptions follow a qualitative order.

4)      A type of exploratory research is described, this type of research is not definitive; Therefore, the investigation conclusions are not absolute, they are preliminary to establish new directly related investigations.

5)      Specify the population enumerated with the symbol of capital N, and the sample studied with a lower-case n, given the existing confusion about these data. If the sample is established in 21 subjects interviewed, as specified in the results section (Line: 344), it is too low to establish generalizations.

6)      Taking into account the interdisciplinarity of the interviewees, it is recommended to specify the particularities of their statements according to their professional specialties.

7)      It is recommended to use percentages to establish statistical trends on the variables used in the interviews.

8)      It is necessary to establish the items or variables investigated in relation to the investigation objective in a concise manner. The design of a table can help to organize and understand the study variables, with the pertinent descriptions.

9)      Specify a subsection in the discussion section named "Strengths and limitations of the research", which characterizes the research, existing in some cases in the paragraphs of the conclusions section.

10)   The conclusions must specify the most relevant findings of the investigation, leaving the recommendations, limitations and investigative strengths in the final paragraphs of the discussion section.

Comments on the Quality of English Language

Consult with an English language specialist

Author Response

We are grateful for all the help provided by the evaluator's opinion. We performed, to the best of our capabilities and understanding, the suggested changes for improvements, in the certainty that there will always remain viewpoints to be considered.

REVIEW FEEDBACK

Dear reviewer,

We are grateful for your relevant suggestions, reviewing the work entitled Older Adults and Fall-Inducing Frailty: Bourdieu’s Analysis of Field Narratives.

ACADEMIC RESPONSES

TITLE:

REVIEW: “There is no evidence of an acceptable relationship between the article title and the investigation objective, specifying the organization of the services.”

POINT-TO-POINT RESPONSES

Reviewer’s request/suggestion  

Responder – Authors' response/adaptations

“There is no evidence of an acceptable relationship between the article title and the investigation objective, specifying the organization of the services.”

We adapted the text.

“It is recommended to specify the improved research objective in the last paragraph of the introduction.”

We highlighted the objective (see lines 184-186).

MATERIAL AND METHODS:

REVIEW: “It is recommended to specify the improved research objective in the last paragraph of the introduction. The research is described as quantitative (Line: 238). However, all the descriptions follow a qualitative order. A type of exploratory research is described. This type of research is not definitive. Therefore, the investigation conclusions are not absolute, they are preliminary to establish new directly related investigations. Specify the population enumerated with the symbol of capital N, and the sample studied with a lower-case n, given the existing confusion about these data. If the sample is established in 21 subjects interviewed, as specified in the results section (Line: 344), it is too low to establish generalizations. Taking into account the interdisciplinarity of the interviewees, it is recommended to specify the particularities of their statements according to their professional specialties. It is recommended to use percentages to establish statistical trends on the variables used in the interviews. It is necessary to establish the items or variables investigated in relation to the investigation objective in a concise manner. The design of a table can help to organize and understand the study variables, with the pertinent descriptions.”

POINT-TO-POINT RESPONSES

Reviewer’s request/suggestion  

Responder – Authors' response/adaptations

The research is described as quantitative (Line: 238). However, all the descriptions follow a qualitative order.”

Thank you for your observation. We apologize for the mistake, which has been corrected (see lines 202-203).

“A type of exploratory research is described. This type of research is not definitive. Therefore, the investigation conclusions are not absolute, they are preliminary to establish new directly related investigations.”

We proceeded with adjustments.

By “exploratory” we understand a phase in qualitative research that precedes the construction of the project, and other phases that follow (the beginning of everything, when the review of the topic is performed in published works to establish the state-of-the-art; first visits are made to the field to recognize and certify the existence of the problem for interested parties; and accessing spaces and identifying informants, ensuring feasibility of data collection, etc.):

When addressing qualitative research, the activities that underpin the exploratory phase precede the construction of the project and often follow it. For example, it is necessary to get closer to the field of observation to better outline other issues, such as the research instruments and the research group. When achieving a broader view, we can say that the construction of the project is even a stage of the exploratory phase.” (MINAYO, Maria Cecília de Souza (org.). Pesquisa Social. Teoria, método e criatividade. 18 ed. Petrópolis: Vozes, 2001.)

“Specify the population enumerated with the symbol of capital N, and the sample studied with a lower-case n, given the existing confusion about these data.”

We respectfully inform you  that the study was not interested in the “sampling fraction” (with N being the number of elements in the population and n being the number of elements in the sample).

“If the sample is established in 21 subjects interviewed, as specified in the results section (Line: 344), it is too low to establish generalizations.”

We appreciate your consideration.

As in any qualitative research, this research does not pursue generalization as quantitative studies – Generalization in qualitative research is analytical-interpretive; in other words, instead of seeking statistical generalization, it focuses on identifying trends and/or patterns , in the themes emerging from the data, and in the elaboration of new concepts to be applied to similar contexts/situations.

In this qualitative research, therefore, instead of a “representative sample”, recruitment aimed at a driving diverse insights within the group/population under study (MINAYO, Maria Cecília de Souza (org.). Pesquisa Social. Teoria, método e criatividade. 18 ed. Petrópolis: Vozes, 2001.)

“Taking into account the interdisciplinarity of the interviewees, it is recommended to specify the particularities of their statements according to their professional specialties.”

Respectfully, we understand that this would result in a new article, with a different focus.

“It is recommended to use percentages to establish statistical trends on the variables used in the interviews.”

Respectfully, we avoid (in qualitative language) statistics as this general guideline to signify the trends' direction.

“It is necessary to establish the items or variables investigated in relation to the investigation objective in a concise manner.”

We did our best, insofar as the study is not interested in establishing an order of values for the nominal qualitative variables.

“The design of a table can help to organize and understand the study variables, with the pertinent descriptions.”

We inserted a table (see lines 284-286)

RESULTS AND DISCUSSION:

REVIEW: “Specify a subsection in the discussion section named "Strengths and limitations of the research", which characterizes the research, existing in some cases in the paragraphs of the conclusions section.”

POINT-TO-POINT RESPONSES

“Specify a subsection in the discussion section named "Strengths and limitations of the research".

“…leave recommendations, limitations, and investigative strengths in the final paragraphs of the discussion section.”

We included this information (see lines 784-788).

Thank you for your guidance (see lines 784-794)..

CONCLUSION:

REVIEW: “ The conclusions must specify the most relevant findings of the investigation, leaving the recommendations, limitations and investigative strengths in the final paragraphs of the discussion section.”

POINT-TO-POINT RESPONSES

“The conclusions must specify the most relevant findings of the investigation.”

We adjusted accordingly.

Comments on the Quality of English Language.

REVIEW: “Consult with an English language specialist.”

POINT-TO-POINT RESPONSES

“Consult with an English language specialist.”

We considered this comment and made a new editing of the article.

Reviewer 3 Report

Comments and Suggestions for Authors

This study is a qualitative study with focus group discussion among 21 health professionals in Brazil. This study provide views on the current trends of ageing and the role of primary health care center in helping to solve the health and society issues related to ageing, particularly falls. There are few comments to help improve the manuscript.

1.  Please better describe the context of fall-inducing frailty in your study as this is one of the main contexts of this study. Falls may cause frailty, but frailty may also lead to falls, and not all older adults who falls are frail. Also, the definition of frail need to be stated clearer. 

2. In the study design part, the authors stated that "It describes part of the results of the thesis entitled Applications of Bourdieu’s epistemology to narratives about longevity and primary health care, developed in a doctorate in the Graduate Program in Health Sciences, Collective Health Concentration, ABC University Center, Santo André-SP, Brazil, and is a self-funded study." Personally I think it is not necessary to mention that this study is part of the thesis as long as this study or the data presented in this manuscript is not publish in the other journal. Consider remove this sentence.

3. The author mentioned that among the 21 participants, 4 were male. However, in page 7, line 350, the author stated that "The fact that only two men participated in the sample suggests a gender bias due to the high degree of female adherence to higher education health courses." Please double confirm how many participants were male.

Author Response

We are grateful for all the help provided by the evaluator's opinion. We performed, to the best of our capabilities and understanding, the suggested changes for improvements, in the certainty that there will always remain viewpoints to be considered.

REVIEW FEEDBACK

Dear reviewer,

We are grateful for your relevant suggestions, reviewing the work entitled Older Adults and Fall-Inducing Frailty: Bourdieu’s Analysis of Field Narratives.

ACADEMIC RESPONSES

INTRODUCTION:

REVIEW: “Please better describe the context of fall-inducing frailty in your study as this is one of the main contexts of this study. Falls may cause frailty, but frailty may also lead to falls, and not all older adults who falls are frail. Also, the definition of frail need to be stated clearer.”

POINT-TO-POINT RESPONSES

Reviewer’s request/suggestion 

Responder – Authors' response/adaptations

“Please better describe the context of fall-inducing frailty in your study.”

We proceeded with adjustments (see lines 58-83)

“...the definition of frail need to be stated clearer.”

We appreciate the guidance. We attempted to “present more clearly” the definition of frail (see lines 63-70)

METHODS:

REVIEW: “ In the study design part, the authors stated that "It describes part of the results of the thesis entitled Applications of Bourdieu’s epistemology to narratives about longevity and primary health care, developed in a doctorate in the Graduate Program in Health Sciences, Collective Health Concentration, ABC University Center, Santo André-SP, Brazil, and is a self-funded study." Personally, I think it is not necessary to mention that this study is part of the thesis as long as this study or the data presented in this manuscript is not publish in the other journal. Consider remove this sentence. The author mentioned that among the 21 participants, 4 were male. However, in page 7, line 350, the author stated that "The fact that only two men participated in the sample suggests a gender bias due to the high degree of female adherence to higher education health courses." Please double confirm how many participants were male.”

POINT-TO-POINT RESPONSES

Reviewer’s request/suggestion 

Responder – Authors' response/adaptations

“...Personally, I think it is not necessary to mention that this study is part of the thesis as long as this study or the data presented in this manuscript is not publish in the other journal. Consider remove this sentence.”

We removed the sentence.

“...in page 7, line 350 …please double confirm how many participants were male.”

We thank you for your kind observation and made the correction (see line 284).

Reviewer 4 Report

Comments and Suggestions for Authors

Dear Authors,

Thank you very much for the opportunity to review the work on "Older Adults and Fall-inducing Frailty: Bourdieu's Analysis of Field Narratives". Below, I have raised several points that the authors should consider and address.

Introduction: While it is meticulously written with concrete details and background justification in the introduction section. Despite this being qualitative research, I felt it may be slightly excessively covered. Some of this information could have been written more succinctly.

Furthermore, more emphasis should be placed on clearly stating the study's objectives.

Methods: Please include information if any software was used for data analysis.

Results: Please consider using a demographic table to present the sociodemographic profile of the 21 health professionals who formed the focus group rather than in plain poses.

Discussion & Conclusion: Similarly to the introduction, I felt that some of the discussion could be streamlined.

In general, much verbiage to describe is akin to stating that "water is wet," as the saying goes. While it is a qualitative study, the overall manuscript could be re-organised to entice the readership of the potential audience. 

Comments on the Quality of English Language

English is well structured but lengthy. 

Author Response

We are grateful for all the help provided by the evaluator's opinion. We performed, to the best of our capabilities and understanding, the suggested changes for improvements, in the certainty that there will always remain viewpoints to be considered.

REVIEW FEEDBACK

Dear reviewer,

We are grateful for your relevant suggestions, reviewing the work entitled Older Adults and Fall-Inducing Frailty: Bourdieu’s Analysis of Field Narratives.

ACADEMIC RESPONSES

INTRODUCTION:

REVIEW: “Introduction: While it is meticulously written with concrete details and background justification in the introduction section. Despite this being qualitative research, I felt it may be slightly excessively covered. Some of this information could have been written more succinctly. Furthermore, more emphasis should be placed on clearly stating the study's objectives.”

POINT-TO-POINT RESPONSES

Reviewer  Request/suggestion

Responder – Authors' response/adaptations

“Despite this being qualitative research, I felt it may be slightly excessively covered.”

We performed adjustments.

“Furthermore, more emphasis should be placed on clearly stating the study's objectives.”

We highlighted them (see lines 184-186)

METHODS:

REVIEW: “Please include information if any software was used for data analysis.”

POINT-TO-POINT RESPONSES

Reviewer  Request/suggestion

Responder – Authors' response/adaptations

“Please include information if any software was used for data analysis.”

No processing software and qualitative analyses were applied to the statements per se. Processing, therefore, was manual, following the narrative method steps informed by research conducted by methodologist Romeu Gomes (references 77-79).

The data produced by the narrative FG interviews required more than 25 hours of transcription. Regarding recording, although we are aware of the availability of software for analyzing narratives (e.g. IRAMUTEQ, freely distributed under the terms of a specific license) and others that organize data (processing), generating analysis categories, we opted to perform verbatim transcription, emerging in the statements and faithfully recording everything that was said, including hesitations, repetitions and other sounds arising from speech defects, but which were not necessarily words – this is how the narrative texts were constructed, as instructed by methodologist Romeu Gomes (corpus).

The speed, precision, stop, and resume editing tools, respecting the iPhone version specific resources were used when we adopted the Literal Transcription to be faithful to the recording.

The quantitative data were processed by entering them into an electronic spreadsheet in the form of a database, for which the Excel® program was used, generating tables as visual resources that helped to present the variables studied, consisting of simple and percentage frequency measurements.

RESULTS:

REVIEW: “Please consider using a demographic table to present the sociodemographic profile of the 21 health professionals who formed the focus group rather than in plain poses.”

POINT-TO-POINT RESPONSES

“Please consider using a demographic table to present the sociodemographic profile of the 21 health professionals who formed the focus group rather than in plain poses.”

Thank you for your guidance. We inserted this quantitative data visualization resource (see lines 284-286).

DISCUSSION AND CONCLUSION:

REVIEW: “Similarly to the introduction, I felt that some of the discussion could be streamlined. In general, much verbiage to describe is akin to stating that "water is wet," as the saying goes. While it is a qualitative study, the overall manuscript could be re-organised to entice the readership of the potential audience.”

POINT-TO-POINT RESPONSES

“…some of the discussion could be streamlined.”

We improved this point.

“In general, much verbiage to describe […]”

We adapted the style to match academic writing.

“…the overall manuscript could be re-organised to entice the readership of the potential audience.”

We dedicated our best endeavors toward reorganizing the manuscript, as suggested.

Comments on the Quality of English Language

REVIEW: “English is well structured but lengthy.”

POINT-TO-POINT RESPONSES

“English is well structured but lengthy.”

We considered this comment and edited the article.

Round 2

Reviewer 2 Report

Comments and Suggestions for Authors

The authors have improved the document content based on the comments issued.

Comments on the Quality of English Language

Assess with a specialist

Reviewer 4 Report

Comments and Suggestions for Authors

Thank you very much for the revision. 

I have no further comments. 

Comments on the Quality of English Language

The quality of English and length of the manuscript has improved.